# Dubai Municipality Initiative to Reduce Food Loss

**Sayed Essam [1,2,\*], Tim Gill [2,3] and Robyn G. Alders [4,5]**

1   School of Veterinary Science, School of Life and Environmental Sciences, University of Sydney, Sydney, NSW 2006, Australia
2   Charles Perkins Centre, University of Sydney, Sydney, NSW 2006, Australia; tim.gill@sydney.edu.au
3   Faculty of Medicine and Health, University of Sydney, Sydney, NSW 2006, Australia
4   Global Health Programme, Chatham House, London SW1Y 4LE, UK; robyn.alders@gmail.com
5   Development Policy Centre, Australian National University, Canberra, ACT 0200, Australia
\*   Correspondence: essssam@gmail.com; Tel.: +97-15-0625-2425

**Abstract:** Dubai has experienced enormous economic and population growth, transforming the city from a small regional business hub in the 1970s to a global business hub of financial and tourism activities in the 21st century. Relevant Dubai Municipality reports were reviewed and semi-structured interviews and focus group discussions conducted with representatives of large food importers and local producers to evaluate the link between the food importation requirements and minimising food loss. Measures taken by the Municipality of Dubai to successfully reduce food loss and improve food security include the diversion of potential food loss to the United Arab Emirates Food Bank and recycling it into animal feed when appropriate. These measures significantly reduced food loss by 93% in the four years from 2016 to 2019. Some political and managerial implications of the study are highlighted.

**Keywords:** food loss; food consignment; food waste; food bank





## 1. Introduction

Although much effort has been directed towards increasing agricultural production to improve food security, food loss has remained a growing concern for many nations in the Gulf region, including the United Arab Emirates. A large amount of food loss is generated at various stages of the food supply chain, including food manufacturing, which negatively impacts food security and the environment [1].

In 2014, the amount of waste documented in Dubai was 494,315 tonnes: 36.8% was dairy waste, 26.3% was from the meat industry, 20.9% was pasta and flour, and 15.8% was seafood waste [2].

As a result of population growth and increasing economic activities, food waste has piled up over the years, contributing significantly to large municipal landfills and dumpsites, which generate large quantities of methane, a potent greenhouse gas. A significant percentage of food waste has been diverted to landfill, and the quantity continues to rise rapidly.

Waste management functions in the UAE Government have been completely devolved to local authorities, which are pursuing critical areas in line with existing regulations using the limited budget awarded to them. This suggests that there is a variation in the approaches employed by different local authorities in dealing with food loss and waste. For instance, in 2008, the Government of Abu Dhabi established the Centre of Waste Management, which is mandated to develop appropriate policies and strategies for managing waste. In 2012, the Dubai Municipality developed an Integrated Waste Management Master Plan that is intended to reduce food waste using integrated and innovative approaches [3]. Moreover, the UAE Government is looking to fulfil the 2030 Agenda for Sustainable Development [4].

In recent years, there have been several attempts to address the problem of the amount of food loss from imported shipments and locally produced food. Developed nations have

adopted numerous methods of managing food waste, such as recycling and transforming waste into clean energy. Los Angeles produces approximately 800,000 tonnes of edible food waste annually [5]. Instead of letting food waste pile up and disposing of it in landfills, New York City has adopted the generation of renewable energy using composting. Other cities have established programs to improve local awareness regarding food waste. Notable examples include the "Love Food, Hate Waste campaign" established in London and "Save the food" based in the United States [6]. An evaluation of the program established in London indicated that there was a 14% reduction in food waste during the first six months. Cities in the UAE have concentrated on reducing its impact on air quality, municipal functions and other waste management functions [7]. Several programs have been initiated to increase public awareness regarding food loss and food waste, and its effects at the national level. The UAE Government recognized the importance of achieving the Sustainable Development Goals (SDGs) of the United Nations (Zero Hunger) by utilizing food and avoiding disposing of it in landfill [4].

Ahmed (2015) reviewed the literature related to food waste, concentrating on establishing the relationship that exists between different available food items, individual eating patterns and food manufacturer operations [2]. A small number of studies examined food waste at the city level. One UAE city that has attempted to address food loss is Dubai, where the government has implemented three main practices, namely, recycling, the development of the UAE Food Bank and the conversion of waste into animal feed. Table 1 presents a description of the specific practices adopted by the Dubai Government in line with the requirements of food safety and the sustainability of the environment.

**Table 1.** Dubai Municipality solutions to reduce food loss [3].

| Practice | Description |
| --- | --- |
| Recycling | Recycling of foodstuffs that are accepted by the recycling companies |
| UAE Food Bank | Acceptance of food complying with the UAE Food Bank standards |
| Animal feed | Accepting food that complies with zoo requirements |

Potential food loss is diverted to the UAE Food Bank, recycling or animal feed, which reduced food loss by 93% in the four years from 2016 to 2019. Recently, the Ministry of Climate Change and Environment (MoCCAE) announced that UAE-based hospitality companies had committed to the challenge of reducing food loss, pledging a reduction equivalent to 1 million meals by the end of 2018. This target was to be increased to 2 million meals in 2019 and three million meals by 2020 [8].

Approximately 90% of Dubai food imports are imported through the ports managed by the Municipality of Dubai. Given the environmental challenges presented by food loss, the Food Safety Department of the Municipality of Dubai launched an initiative in 2016 to reduce the losses caused by the rejection of shipments due to incorrect documentation, food not meeting the importation requirements or specifications or food found to be unfit for human consumption, after testing samples.

Before implementing the initiative to reduce the food loss in Dubai, the Dubai Municipality did not provide alternative solutions for food-importing companies before taking the food shipments to landfill. Therefore, food companies would take the food to landfill. The downfall of the procedure of directing food companies towards alternative solutions, such as recycling companies, food banks and animal farms, is that landfill is an easier solution for food companies to dispose of extra food.

The aim of the current study is to evaluate the link between importation requirements as a measure for monitoring and minimising food loss and meeting the UAE 2030 Agenda for Sustainable Development. Specific questions answered by this study include: What are the main causes of food loss in Dubai? What innovations have helped to reduce food loss in Dubai? Additionally, which of these may be applicable within the present system of food distribution to reduce food loss in Dubai? To achieve this, a strategic analysis of the

food supply chain in Dubai was performed to identify causes of food loss, and a systematic exploration of innovations that may help reduce food loss was conducted.

The scope of the study covered only the food loss part of the value chain because no such studies have been conducted in this part in Dubai. The study did not include food waste at the level of households, retailers, catering companies and hotels. This is attributed partly to the fact that the Waste Management Department in Dubai Municipality has not yet devised any mechanism they can use to segregate food waste from agricultural waste. The Waste Department in Dubai Municipality has played a critical role in addressing increasing food waste in a number of ways, including: (i) receiving and reviewing food destruction requests from local food manufacturers or food importation companies; (ii) supervising the destruction of large quantities of food pending the decision of the Dubai Municipality regarding its suitability for human consumption; and (iii) computing the amount of food required to be destroyed and separating it from other products. On the other hand, the Waste Department is charged with the mandate of dealing with the disposal of food at the household, retail or company level. The Waste Department is also responsible for dealing with agricultural waste, which accounted for approximately 17% of the aggregate waste in Dubai in 2020 [9].

For the purpose of this paper, food loss and food waste are defined as follows:

Food loss refers to a decrease, at all stages of the food chain prior to the consumer level, in the mass of food that was originally intended for human consumption, regardless of the cause.

Food waste refers to food appropriate for human consumption being discarded or left to spoil at the consumer level, regardless of the cause [10].

*Food Destruction Mechanism before Implementing the Food Loss Initiative*

Before 2016, food companies applied to the Dubai Municipality Food Department to obtain initial approval for destruction of food past its expiry date or not meeting the importation requirements. After receiving approval, the food companies took this approval to the Dubai Municipality Waste Department and applied for diversion of the food to the main landfill of Dubai city. When food to be destroyed arrived at the landfill area, it was weighed and buried in the landfill under the supervision of both the Food Safety Department and Waste Department to make sure the food did not return to the market [3].

## 2. Methodology

### 2.1. Study Area

The UAE, in the southern part of the Arabian Peninsula, has a population of over 9 million, with one of the highest per capita incomes in the world. This includes an aggregate of 180 different nationalities, all within the boundaries of the UAE, regardless of their citizenship status. Based on information provided by FAO et al., 2018, South Asians were the largest group of non-nationals in UAE, representing 58%, closely followed by other Asians (17%) and Western expatriates (8.5%) [11]. The population of Dubai grew from 183,000 in 1975 to 3.36 million in 2019 [12]. This has been coupled with increased investment and development projects, which have spurred economic and population growth.

### 2.2. Data Collection

#### 2.2.1. Analysing Existing Data

Data for this study were drawn from the Dubai Municipality, UAE Food Bank and interviews with key officials of food importation companies. Data collected from the Dubai Municipality Food Safety Department consisted of quantities of food destroyed, recycled and converted into animal feed. Quantities of food destroyed by food importation companies during the period 2015–2018 were also solicited from available secondary data in terms of food production, security and loss.

Two research instruments were used to collect data in the study, and each is discussed in the subsequent section. First, a questionnaire (Supplementary Materials S1) was admin-

istered to five key officials in the Food Safety Department and Waste Department of Dubai Municipality and six food companies in September 2018. The companies were selected based on their products, namely, two chicken companies, one dairy company and three food retailers. In order to obtain an accurate measure of food loss, the study compared the mass of food meant for human consumption and lost through supply, using available data and the global food loss literature.

### 2.2.2. Qualitative Data

Framework analysis was conducted, and key officials in the Dubai Municipality Food Safety Department and selected food importation companies in the Emirate of Dubai were interviewed. Food importation companies were selected based on criteria provided by the Dubai Municipality Food Safety Department (FSD). The researchers also conducted focused group interviews with the officials from the Food Safety Department in Dubai Municipality. The interviews concentrated on methods used to reduce food loss, initiatives for improving food security and existing support for increased production by local farms. An interview guide is provided in Supplementary Materials S2 covering different aspects associated with food export and import, food distribution and loss in different cities in UAE and the ability of the company to contribute to sustainable food security. Table 2 presents the details of the participants and the number of interviews conducted.

**Table 2.** The details of the participants and number of interviews conducted within each group.

| Title of the Interview | Number of Interviews | Target Group | Dubai Municipality | Supervisors | Researchers | Total Attendees |
|---|---|---|---|---|---|---|
| Food suppliers | 2 | 4 suppliers from 3 companies / 2 retailers from 2 companies | 2 staff from Food Safety Department | 2 | 1 | Interview 1: 9 Interview 2: 7 (6 from target group + 2 FSD) |
| Focus group discussion | 2 | 2 specialists from Food Safety Department / 1 expert from Food Safety Department | | 1 | 1 | Interview 1: 4 Interview 2: 3 (3 from target group) |
| Food Safety Department | 1 | 4 staff from Food Safety Department | | 2 | 1 | 7 (4 from target group) |

Interviews were conducted in English, and, in some cases, the researchers translated interview questions to Arabic to make sure all the attendees understood them. The interviews lasted for approximately 45–60 min and were carried out in a meeting room.

### 2.2.3. Data Analysis

In the qualitative phase, an in-depth analysis of the data collected from the interviews and focus groups was conducted to identify the emerging aspects. The questions were structured to make it easier to collect data from the interviews and focus group discussions. Emerging themes and sub-themes from the interviews and focus groups were identified through codes in order to understand the food destruction mechanism before and after implementing the Food Loss Initiative and to understand the challenges that were faced by the food companies when the process was changed.

### 2.2.4. Ethical Approval

Ethical approval was granted by the University of Sydney, Human Research Ethics Committee (2018/033), as illustrated in Supplementary Materials S3. All respondents signed the Participant Consent Form (Supplementary Materials S2) and were assured of confidentiality, anonymity and voluntary participation. This study discusses challenges faced by specific food department entities and food companies. Therefore, the Dubai

Municipality Food Safety Department and food companies were assigned unique identifier codes to maintain commercial confidentiality.

## 3. Results

This section presents the results obtained from the analysis of both qualitative and quantitative data gathered from key officials in the Dubai Municipality Food Safety Department and selected food importation companies.

From the interviews, the researchers identified key areas of food loss from all participants: Food Safety Department, focus group and food companies.

The study critically analysed the emerging themes that repeatedly appeared in the participants' responses to the open-ended inquiries regarding the Food Loss Initiative. A thematic analysis of the interviews and focus group discussions revealed emerging themes focusing on the Food Loss Initiative among the different groups interviewed, namely, the shelf life of the food product, regulation, food loss, food bank and food waste. There was a general consensus on the need for a better solution to avoid food destruction, including updated procedures towards diverting rejected food shipments from the landfill path. Table 3 presents all the emerging themes across the different groups and their frequency of occurrence.

**Table 3.** The details of the emerging themes across the different groups.

| | | Title of the Interview | | |
|---|---|---|---|---|
| **Codes** | **Themes** | **Food Suppliers** | **Focus Group Discussion** | **Food Safety Department** |
| UAE/Dubai Government | Food Bank | X | | X |
| | Regulation of importation | X | X | X |
| Challenges | Product shelf life | X | X | |
| | Food loss | X | | X |
| | Food waste | X | X | |

A comprehensive thematic analysis of the three identified themes was achieved through the examination of the participants' background and their responses to the research questions of this study. The participants' views concerning the environment and logistics formed the basis of identifying possible pathways for improving the initiative.

The food companies and manufacturers were mainly concerned about the regulation of shelf life and the challenges influencing food importation.

The Food Safety Department held a similar view concerning the need for stringent regulation; however, the official hinted that the shelf life of fresh products should be flexible to give food companies a chance to keep their products on the shelf for longer and to avoid food destruction. One company official proposed that "The UAE standard for mandatory shelf life should be revised to be able to keep the food on the shelfs for longer and reduce the food waste, especially for the fresh product".

Concerning the Food Loss Initiative, there were several observations that emerged from the interviews. First, there was a general consensus that the initiative helped to reduce food loss. Second, it was noted that the shelf life of the product is among the major challenges that should be revised. Third, the UAE Food Bank is the best solution for food nearing its expiry date.

### 3.1. Participating Food Safety Department Characteristics

The Food Safety Department is mandated to develop and implement critical procedures that can reduce food loss in a cost-effective manner. Food companies are required to seek approval from the Food Safety Department through an online application process for food destruction. The examination of the data indicated that, in 2016, the Food Safety

Department received an average of 59 requests per day from food importation companies seeking permission to dispose of food that did not comply with food importation regulations. In 2019, the Food Safety Department received an average of two requests per day from food importers and local producers to destroy food.

### 3.2. Participating Company Characteristics

Food companies were interviewed about food loss issues. The food companies were divided into categories A to G based on their products as follows: dairy (A), imported chickens (from Brazil) and meat cuts (B), local meat and chicken production (C), retail foods (D and E), ready-made meals (F) and camel milk production and supply (G).

### 3.3. Food Imports

The Dubai Government has enacted various regulations in an effort to limit the importation of foodstuffs and minimise food losses. A comparison of food imports during the period 2015–2018 is provided in Figure 1 below. The results indicate that there was a reduction in food imports from 8650.368 tonnes in 2017 to 7505.337 tonnes in 2018. The number of food shipments remained relatively constant from 2015 to 2018.

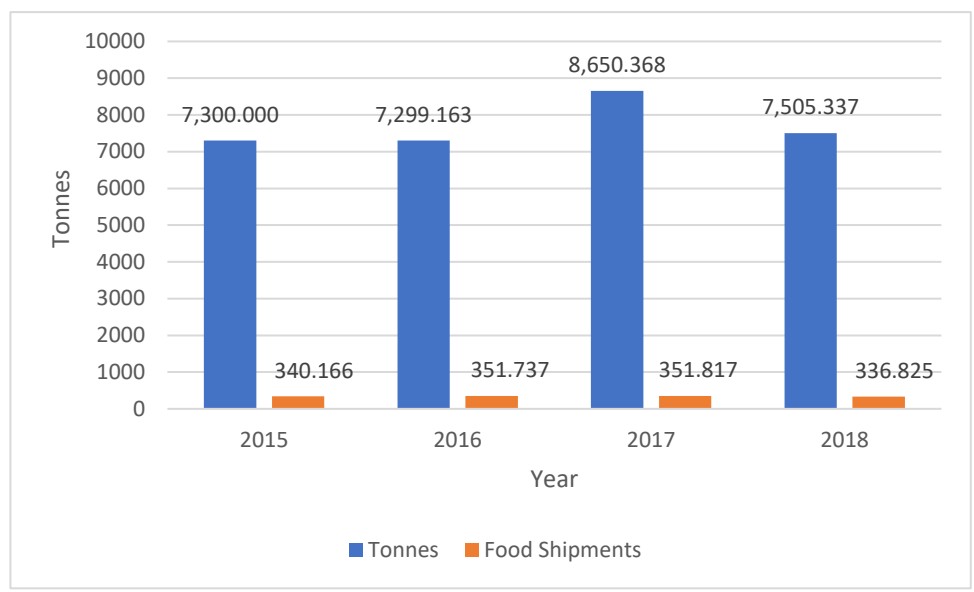

**Figure 1.** Food imports and shipments in UAE during the period 2015–2018 [3].

The statistics shown in Figure 1 are all the food imports to Dubai via sea, air and land ports. The total number of entering points for the ports is 10. The sea ports have three entering points, airports have two entering ports, and the land ports have only one entry point for food.

### 3.4. Food Loss in Dubai Municipality

Figure 2 presents the reduction in food loss in the Dubai Municipality during the period 2015–2018. The results show that there was a continuous reduction in food loss in the Dubai Municipality, from 16,909 tonnes of food in 2015 to 1100 tonnes in 2018. The significant decline in the volume of food destroyed was attributed to an initiative launched by the Food Safety Department early in 2016 to reduce food rejected due to non-compliance with documentation requirements, not meeting the importation requirements or specifications, or food found unfit for human consumption, after testing samples. The initiative included alternative solutions to the process of destroying food, including donations to the UAE Food Bank, recycling and conversion to animal feed. The initiative also worked to raise the awareness of food companies.

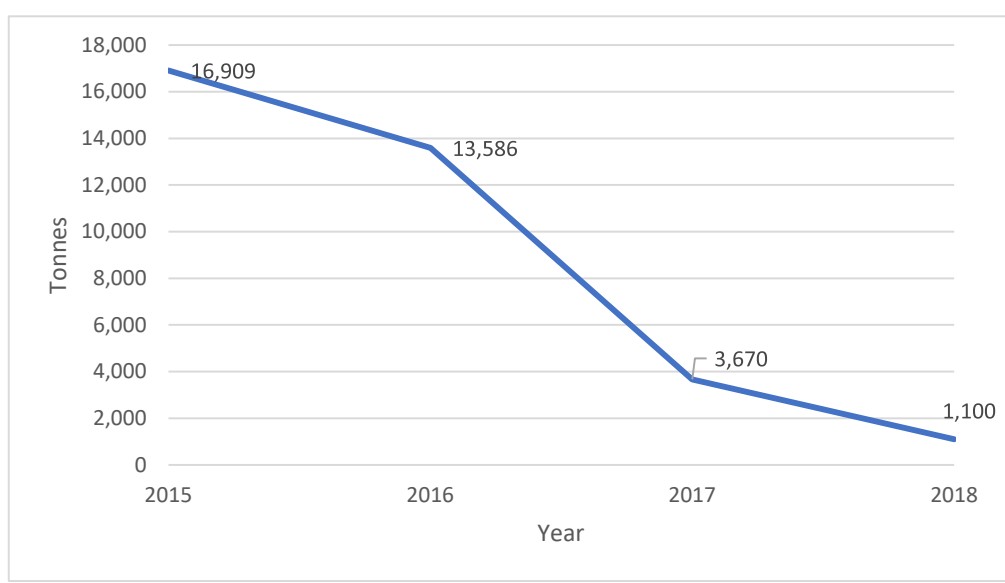

**Figure 2.** Reduction in food loss in Dubai from 2015 to 2018 [3].

Since the Dubai Municipality food loss initiative was launched in 2016, a total of 8423 tonnes of food have been diverted from landfill. In 2018, recycling assisted in the diversion of approximately 5428 tonnes (64%), 2315 tonnes (27%) were diverted to the UAE Food Bank after complying with standards for human consumption, and 679 tonnes (8%) were diverted to animal feed (Figure 3).

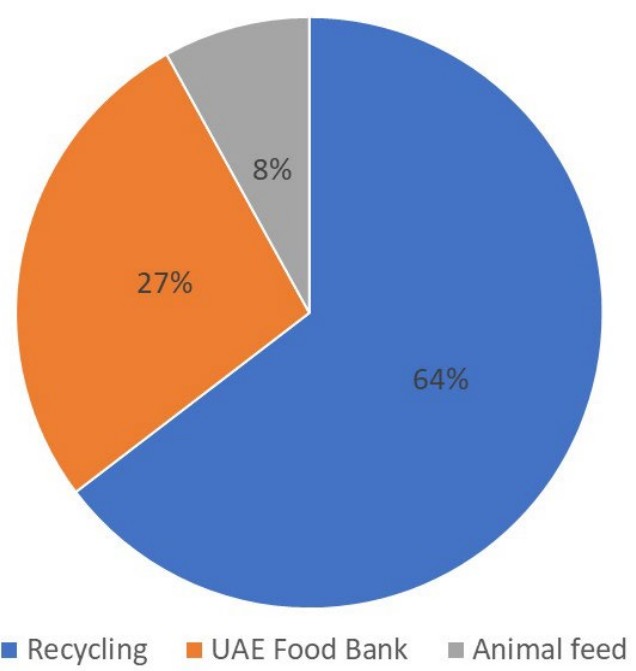

**Figure 3.** Outcome of food diverted from destruction to alternative solutions in 2018 [3].

All three solutions were conducted under the supervision of the Dubai Municipality Food safety department to ensure the food standards were reached and did not go back to market without approval.

*3.5. Food Destruction Mechanism after Implementing the Food Loss Initiative*

After implementing the initiative, the procedure for food destruction completely changed. The Dubai Municipality Food Safety Department and Waste Department selected recycling companies that were able to convert waste foodstuffs to compost and recycle the bottles, cans and glass containers of the foodstuffs. They also met with the Head of the Ras Al Khaimah Zoo to ensure that they were able to receive meat that has documentation problems and is suitable for the carnivores in the zoo. Finally, they met with the UAE Food Bank staff to make sure they were also able to receive food that does not meet documentation and labelling requirements, to take samples before distribution and distribute to people in need. After announcing the initiative, the Dubai Municipality Food Safety Department gave a grace period for the companies to change their policy and procedures for food destruction and to look for alternative solutions.

Food that is not suitable for human or animal consumption is directed to recycling companies. The recycling companies receive and process the foodstuff and send a report to the Food Safety Department and Waste Department in the Dubai Municipality. Similarly, the UAE Food Bank and zoo receive and process the food and send reports showing the quantity received to the Food Safety Department and the Waste Department. Food safety inspectors randomly inspect the recycling companies, zoo and the UAE Food Bank to make sure the process is going well. The reports for each of the three solutions go to the Food Safety Department in Dubai Municipality each month.

A reduction in food destruction was noticed immediately after the initiative was announced. In 2017, the Food Safety Department of Dubai Municipality stopped accepting destruction requests from food companies and only accepted requests after revision and the evaluation of the reason for not complying with the regulation. This reduced the destruction of food from 13,586 tonnes in 2016 to 3670 tonnes in 2017. More restrictions were introduced in 2018, and this reduced food destruction to 1100 tonnes.

3.5.1. Positive Actions and Feedback from Key Stakeholders

Data collected from key officials from Dubai Municipality were reviewed using thematic analysis, and a number of emerging themes were noted. Major themes observed from the examination of semi-structured interviews include cost reduction and recycling. Companies A, C, D, E and F all reported reduced food loss after the introduction of the initiative in 2016.

Company A (dairy company) reported achieving a sharp drop in dairy food loss, from nine to two tonnes per day in 2016, and indicated that this also reduced the costs of food dumping and saved time. Company C (meat and chicken manufacturer), a local manufacturer of food products, reported a reduction in food loss by tracking expiry dates, scheduling appropriate distribution and sending products approaching their expiry dates to the UAE Food Bank and labour settlements. Importing Company D (food retailer) reduced its loss of chicken products by distributing products nearing their expiry dates to settlements for foreign workers, after processing and cooking raw foodstuffs. Retail Companies E and F reported reduced food destruction by organising on-demand delivery from the market, which helped reduce the cost of storage.

Company F (food retailer) mainly imports food in bulk quantities through seaports, as raw materials are required. Air freight is only used for samples and non-commercial consignments. There is a huge market for exporting food to Saudi Arabia and other GCC countries due to the very large population. Frying oil is purchased from the local industry to promote food security in the region. As the production of the company's products purely depends on the purchase order plan from the sales department, it is very rare for the company to send food for destruction. The trading department of the company has trade agreements with its customers involving long-term contracts for product sales. The company has a quality and safety department to check the standard of the imported raw materials and the finished products and to meet regulatory requirements.

Company A (dairy company) was very satisfied with the policies of the Dubai Government, especially the Dubai Municipality, for their focus on customer satisfaction. The company specifically mentioned the food trade Person in Charge (PIC) program of the Food Safety Department, which helps all consignees to better understand food regulations and departmental policies on the inspection and subsequent release of imported consignments. The company agreed that they were able to reduce food loss drastically in 2018 compared to 2016.

### 3.5.2. Challenges Faced by Stakeholders in Reducing Food Loss Based on the Outcome of the Interviews
Regulatory Environment

A representative from Company B (chicken company) reported that a regulatory grey zone in the wording of halal slaughter standards issued by Emirates Standardization and Metrology and Dubai Municipality created uncertainty about the halal status for meat and poultry importers. Food regulation from the Cooperation Council for Arab States of the Gulf (GCC) differs from the UAE regulation, which has a negative influence on the import and export businesses in the Gulf region. However, Dubai regulatory bodies are perceived as being more cooperative and flexible and can solve issues to a certain extent.

Companies also face major challenges due to the regulatory restriction on the shelf life of products, especially fermented and flavoured yoghurts, and would benefit from the shelf life being extended. Imported products last longer than the allowed shelf life, and this may affect the company's marketing strategy. It was noted that the Emirates Authority for Standardization and Metrology should also consider relaxing standards on food additives, as the Codex system is not the only standard used worldwide. Company E (food retailer) made several requests for an extension of the shelf life of certain short-shelf-life dairy products, such as fresh milk and yoghurt, to overcome the food loss situation. Company F (food retailer) also expressed the need for the shelf life of some food commodities to be re-examined and extensions given for those products that have been given unnecessarily short shelf lives by local authorities.

Company E (food retailer) urged the regulatory body to make the printing of the production date on food labels a non-mandatory requirement but also supported and emphasised collaboration between the Gulf countries by following uniform food regulations, food standards and requirements.

Company F (food retailer) believed that small traders are affected by the frequent changes in regulations by Emirates Standardization and Metrology. This company mentioned a lack of consistency between government departments in uniformly applying food regulations.

Company F also requested the authorities to be flexible on consignment-related supporting certificates (such as organic and GMO-free certificates) if they were not available during inspection and asked the authorities not to penalise the company for this reason.

Market-Related Issues

Company G (dairy company) expressed dissatisfaction with the lack of demand for camel milk and its products. The company suggested that reasons for low demand from consumers in Dubai could include the high prices (three to four times more expensive than cow milk); cultural barriers; seasonal production (camels do not produce milk throughout the year as the calf and camel cow are kept together for milking); limited production, with only 6–7 L of milk per camel compared to 40 litres per cow; and a lack of research on animal feed requirements, which influences milk production in camels and is known to vary for different species and breeds [13].

A representative from Company A (dairy company) suggested that there is need for more research to improve the quality of their products and launch new products that meet the needs and interest of their customers. Unfortunately, no market data are available to the company on the age group of their customers. The company's R&D team is working

to develop value-added products. The company's strategic plan includes placing more emphasis on producing fortified products in all food commodities, including dairy and juices, in the future.

Food Quality

Other challenging factors for food storage and transport are the variable and transient populations in the city. Company F dedicated more attention to its premium customers to maintain the quality of the product and ensure punctuality in the delivery of goods. Company F indicated that 30–35% of premium sale products are re-directed to the local market where the percentage of profit is lower.

Maintaining the quality of meat products was also considered a challenge as it affected the cost of finished products. For example, Company B had 16% protein in its chicken sausage product, but local manufacturers had only 13% protein.

Supportive Environment

Company E indicated that it would like more support through subsidies from the Government sector to support its business. The federal government provided initial financial support to Company E for the establishment of the food loss initiative because it was a new concept, but support was later reduced.

## 4. Discussion

Food loss is a major challenge facing the region, and the UAE in particular, and it has various causes. The main contribution of this study is the exploration of the role of food importing and manufacturing companies in the Emirate of Dubai in food loss. The results indicate that the Municipality of Dubai's food loss initiative has achieved positive results that can be followed by other jurisdictions in other areas with similar conditions. The initiative also has a humanitarian benefit through the amount of food transferred to beneficiaries through the UAE Food Bank, and it helped the country direct itself towards the SDG of Zero Hunger mainly through the UAE Food Bank [7].

Comparisons can be made between the Municipality of Dubai and the Singapore National Environment Agency. The increased population in Singapore and the growth of the economy resulted in an increase in food wastage of up to 30% over the last decade. Singapore National Environment Agency set a goal to become a Zero Waste Nation in 2019 with a 30% domestic recycling rate by 2030 [14]. Efforts have been made to prevent and reduce food wastage at the consumer level by developing a hierarchy of food waste to increase public awareness (Figure 4). Their efforts to promote food recycling increased the recycling rate from 13% (76,700 tonnes of food waste recycled; 529,400 tonnes of food waste wasted) in 2009 to 17% (126,200 tonnes of food waste recycled; 636,900 tonnes wasted) in 2018 [9].

The major difference between the Dubai Municipality food loss initiative and the Singapore National Environment Agency program is that the Dubai Municipality initiative focuses on imported food shipments, while the Singapore program focuses on market food handling. In 2016, the Dubai Municipality introduced legislation to reject shipments that do not comply with the importation requirement upon arrival, enabling the Dubai Municipality to achieve remarkable results within a relatively short period.

Despite the interesting results of our study, there is still much research to be conducted.

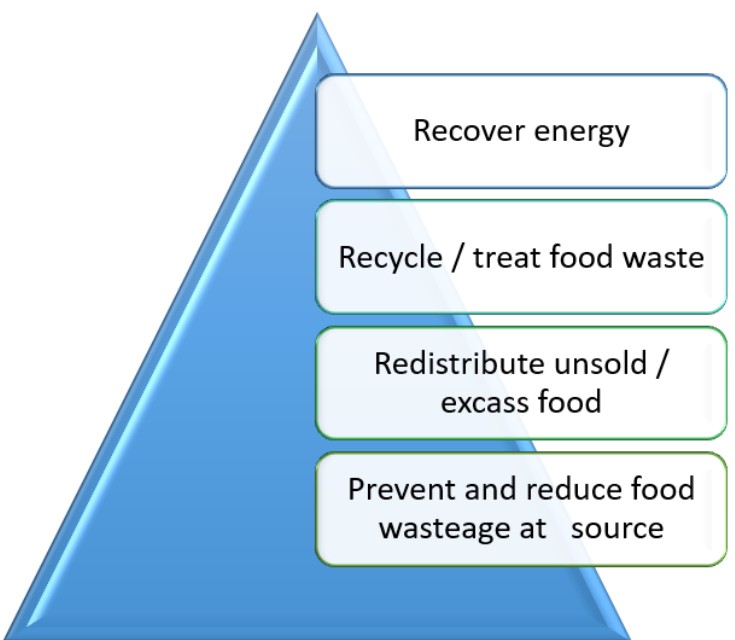

**Figure 4.** Singaporean Environmental Protection Agency food recovery hierarchy [9].

First, it is necessary to ensure that a consistent definition of food loss and uniform methods of calculating the amount of food loss are used in the UAE in order to collect direct data on food loss (i.e., the actual weight of food loss not subject to reduction) and accurately compare the data across the UAE. In addition, indirect surveys can support direct data collection by focusing only on food loss. Surveys can collect additional variables, such as net weight or liquid quantity, that are not included in our data set, which is a constraint on our research. Since there is a link between environmentally friendly industrial production and food loss at the national level, it is important to further explore this relationship and involve the public and private sectors in the implementation of intervention policies. Sawaya (2017) encouraged multi-stakeholder collaboration, especially public–private partnerships, to reduce food loss [15].

Second, official surveys should be based on representative samples, which will enable researchers to analyse and compare data at the sub-national level, such as municipalities, because the local environment is a critical dimension for policy makers when planning interventions. Local policy makers can play a crucial role in reducing the amount of food loss. The authors strongly recommend that local policy makers invest in community interventions to reduce food waste.

Looking into having more studies to increase the knowledge of food manufacturing and to use new technologies, such as food irradiation, to increase the food shelf life in order to reduce the food loss and waste is important [16].

Despite the widespread awareness of the benefits of a healthy diet, at the individual or household level, people do not seem to be fully aware of the consequences of uneaten food on the natural, social and economic environments in which they live. Thus, the issue of food waste at the household level is an important emerging research area for social scientists who could investigate ways to raise awareness of individuals about the environmental, economic and social impact and reduce the level of food waste by individuals. More research is needed on which concepts of food waste can be attributed to differences in the behaviour of individuals towards food loss compared to other countries and which can be attributed to socio-economic and environmental differences and the industrial characteristics of the area where individuals live. Ahmed (2015) stated that awareness campaigns alone could be ineffective because individuals act differently from their presumed habits. The "cascading training" proposed in their study included a group of people that were first trained and then trained others, which could be helpful in reducing food loss. Awareness of the importance

of reducing urban transport emissions and using sustainable vehicles is important during food transportation. Encouraging food companies to find sustainable vehicles will add a new value to this sector and will reduce the emissions caused by food transportation [17].

**5. Conclusions**

Reducing food loss contributes to achieving the one health approach and the UAE 2030 Agenda for Sustainable Development. Utilizing potential food loss by giving it to people who are in need, recycling and converting it into animal feed helps the environment by reducing the food that goes to landfill. Food loss poses a serious challenge not only to the city of Dubai but also to the UAE and the world. This paper identifies the importance of collaboration between companies that import or manufacture food, regulatory bodies and policy makers to successfully address the challenge of food loss. The food loss reduction initiative in Dubai has clearly reduced the proportion of food lost and has developed solutions to optimise food use, including environmentally friendly options, thereby working towards achieving the Sustainable Development goals. Food importation companies have benefitted from this initiative by reducing the food imported to the UAE while still meeting the needs of the local market. The decrease in food loss and optimisation of food use also has a humanitarian aspect, with substantial food transfers to beneficiaries through the UAE Food Bank seen during this initiative.

This study provides political implications; thus, Dubai's initiative to reduce food loss is achieving very positive results, and the approach can be followed in other jurisdictions with similar conditions. Awareness programs could help to achieve the implementation of a similar initiative based on the internal situation of the other Emirates or neighbouring countries.

Additionally, this study provides theoretical and managerial implications [18]. The theoretical implications could be expanded to include restaurants, hotels and other food producers where food is wasted in large quantities.

Managerial implications for the other Emirates or neighbouring countries could be implemented by diagnosing the amount of food loss in their Emirates or neighbouring countries and starting to implement the initiative in a way that suits the local regulation and procedure. Having a clear action plan will help in achieving the target of reducing food loss.

**Supplementary Materials:** The following supporting information can be downloaded at: https://www.mdpi.com/article/10.3390/su14095374/s1, Supplementary Materials S1–S3.

**Author Contributions:** Conceptualization, S.E.; methodology, S.E.; software, Word; validation, T.G. and R.G.A.; formal analysis, S.E.; investigation, S.E.; resources, S.E.; writing—original draft preparation, S.E.; supervision, T.G.; project administration, S.E.; funding acquisition, S.E. All authors have read and agreed to the published version of the manuscript.

**Funding:** This research received no external funding.

**Institutional Review Board Statement:** This study was conducted according to the guidelines of the Declaration of the University of Sydney and approved by the Human Research Ethics Committee (2018/033) on 22 March 2018.

**Informed Consent Statement:** Informed consent was obtained from all the participants. The study aims and the voluntary nature of participation were explained to all the participants before enrolment in the study. Each participant signed the participation form before collecting the data.

**Acknowledgments:** The authors would like to thank the participating companies for their willingness to contribute to this study and the Food Safety Department and Veterinary Service Section in Dubai Municipality, food companies, food retailers, food manufacturers, animal farmers, animal feed retailers and animal feed manufacturers for their support and for providing information. Also I would like to thank Mary Young for editing this manuscript.

**Conflicts of Interest:** The authors declare no conflict of interest.

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
