# Peer review of "Dubai Municipality Initiative to Reduce Food Loss"

_sustainability, doi:10.3390/su14095374_

Round 1

Reviewer 1 Report

Revision process of the manuscript "Dubai Municipality Initiative to Reduce Food Loss (2016–2019) "has been finish. The paper is excellent and deals with important points. And I accept this paper for publication after minor comments

  • Why haven't other places with high food loss, such as wedding halls, hotels and hospitals, been studied? I suggest in future work to collect data from these places
  • Please revise English language with support of a native speaker.

Author Response

Thanks a lot for your valuable comment. Please find my reply to your valuable comments .

Reviewer 2 Report

Thank you for the opportunity to read this interesting article aimed to analyze the Dubai Municipality Initiative to Reduce Food Loss (2016–2019). The topic of the paper is relevant and original in respect to the field of application. However, the quality of the paper could be improved to give adequate value to the research. Specific comments are reported below.
I suggest to improve the section Introduction by better supporting the gap in the literature review on this topic and the originality of the manuscript. Also the link with the Sustainable Development Goals of the United Nations must be included and discuessed. 
The end section must be significantly improved with a specific focus on the managerial, political and theoretical implications of the study. 

In order to improve the paper interesting sources are:

- to reduce food loss:
Galati, A., Moavero, P., & Crescimanno, M. (2019). Consumer awareness and acceptance of irradiated foods: the case of Italian consumers. British Food Journal.
Parlato, A., Giacomarra, M., Galati, A., & Crescimanno, M. (2014). ISO 14470: 2011 and EU legislative background on food irradiation technology: The Italian attitude. Trends in food science & technology38(1), 60-74.

- About food loss:
Fiore, M., Chiara, F., & Adamashvili, N. (2019). Food Loss and Waste, a global responsibility?!. Food Loss and Waste, a global responsibility?!, 825-846.
Fiore, M. (2020). Food loss and waste: the new buzzwords. Exploring an evocative holistic 4Es model for firms and consumers. EuroMed Journal of Business.

- Policy to reduce food loss:
Galati, A., Crescimanno, M., Vrontis, D., & Siggia, D. (2020). Contribution to the sustainability challenges of the food-delivery sector: Finding from the deliveroo italy case study. Sustainability12(17), 7045.

Author Response

Thanks a lot for your valuable comments, please find my reply

Reviewer 3 Report

Observations and recommendations for the article

“Dubai Municipality Initiative to Reduce Food Loss (2016–2019)”

  • “over the three years from 2015 to 2018.”, p. 1, p. 2  - four years! In the title is specified (2016-2019)

  • Recommendation: delete “(2016-2019)” from the title of the article.

  • “As a result of increasing population growth and economic activities, …” - As a result of population growth and increasing economic activities,…"

  • Missing the explanation for “FSD” in Table 2, p. 4

  • In Figure 1, there is no data for 2015; then eliminate this year from this chart, and specify 2016-2018 in

“Figure 1. Food imports and shipments in UAE during the period 2015–2018.”

  • It is missing the unit measure on OY axis of the chart of Figure 1?!

  • The entity text 3.3 is ending with the chart. Write at least one sentence or phrase after the chart, before starting subchapter 3.4.

  • “continuous reduction in food loss ……., from 16,909 tonnes of food in 2015 to 1100 tonnes in 2018.” – use separator of three digits at 1,100. Also at p. 6 “a total of 8423 tonnes….. Recycling assisted in the diversion of approximately 5428 tonnes, … 2315 tonnes…”. Also at p. 7, “This reduced the destruction of food from 13,586 tonnes in 2016 to 3670 tonnes in 2017. More restrictions were introduced in 2018, and this reduced food destruction to 1100”

  • Rebuilt the chart from Figure 2!!!! Why did you let Series1 at each point!?? Why don’t you put the years an Ox axis?!

  • Specify the period for the data in Figure 3. Outcome of food diverted from destruction to alternative solutions.

  • The entity text 3.4 is ending with the chart of Figure 3. Write at least one sentence or phrase after the chart, before starting subchapter 3.5.

  • “Comparisons can be made between the Municipality of Dubai and the country of Singapore.”!! There are no comparisons!

The article has a too describing character, without any original contribution! It doesn’t present interest for any group of readers. The methodology doesn’t refer to the qualitative and quantitative research, the data processing is missing. The establish period in the title cannot be retrieved in the content of article. 

Reviewer

Author Response

Thanks a lot for your valuable comments , please find my reply

Reviewer 4 Report

The manuscript is interesting, however, does not present a holistic vision and specific data from the food waste from this country, the origin of food, lost in the chain of value and real options for a) increase local production, b) reduce lost, c) Uses for sustainable handling of food waste. The manuscript lacks a systematic review or a transdisciplinary review of the complex and widely investigated theme of food waste. Therefore, consequently its theoretical framework, background, discussion of results and conclusions are very poor. The manuscript must incorporate the statistical analysis of the data and incorporate into all sections the objectives of the 2030 Agenda of Sustainable Development

Author Response

Thank you very much for your valuable comments. Please find my reply 

Round 2

Reviewer 3 Report

New recommendations for the article

“Dubai Municipality Initiative to Reduce Food Loss"

There are two subchapters numbered 2.2.4. the first should be numbered 2.2.3 Data Analysis.

Prior to this subchapter 2.2.3, under Table 2, there must be at least one sentence (phrase).

Please change Figure 2 to have the years on the OX axis! I previously recommended that you rebuild this diagram! It is a diagram of the evolution of an indicator, nothing else! It is very annoying to see at each point of the graph repeating the word "year" .... If you want, you can write "year" once as the title of the Ox axis, in the lower right corner or below the diagram, in the middle. It is about an evolution, so take the diagram with lines, without rounded lines!

Reviewer

Author Response

Thanks For your valuable comments, please find my reply.

Reviewer 4 Report

The previous recommendations have not been incorporated in full. Mostly, the manuscript must incorporate  into all sections the objectives of the 2030 Agenda of Sustainable Development

Author Response

(The authors gave the same response as above.)
